# Early Respiratory Physiotherapy versus an Individualized Postural Care Program for Reducing Mechanical Ventilation in Preterm Infants: A Randomised Controlled Trial

**DOI:** 10.3390/children10111761

**Published:** 2023-10-30

**Authors:** Milena Tana, Anthea Bottoni, Francesco Cota, Patrizia Papacci, Alessia Di Polito, Arianna Del Vecchio, Anna Laura Vento, Benedetta Campagnola, Sefora Celona, Laura Cricenti, Ilaria Bastoni, Chiara Tirone, Claudia Aurilia, Alessandra Lio, Angela Paladini, Stefano Nobile, Alessandro Perri, Annamaria Sbordone, Alice Esposito, Simona Fattore, Paola Emilia Ferrara, Gianpaolo Ronconi, Giovanni Vento

**Affiliations:** 1Unità Operativa Complessa di Neonatologia, Fondazione Policlinico Universitario A. Gemelli IRCCS, 00168 Rome, Italy; milena.tana@policlinicogemelli.it (M.T.); anthea.bottoni@policlinicogemelli.it (A.B.); francesco.cota@policlinicogemelli.it (F.C.); patrizia.papacci@policlinicogemelli.it (P.P.); chiara.tirone@policlinicogemelli.it (C.T.); claudia.aurilia@policlinicogemelli.it (C.A.); alessandra.lio@policlinicogemelli.it (A.L.); angela.paladini@policlinicogemelli.it (A.P.); stefano.nobile@policlinicogemelli.it (S.N.); alessandro.perri@policlinicogemelli.it (A.P.); annamaria.sbordone@guest.policlinicogemelli.it (A.S.); alice.esposito@tiscali.it (A.E.); simona.fattore@guest.policlinicogemelli.it (S.F.); 2Servizio Medicina Fisica e Riabilitazione, Fondazione Policlinico Universitario A. Gemelli IRCCS, 00168 Rome, Italy; alessia.dipolito@policlinicogemelli.it (A.D.P.); arianna.delvecchio@policlinicogemelli.it (A.D.V.); annalaura.vento1@policlinicogemelli.it (A.L.V.); b.campagnola@libero.it (B.C.); sefora.celona@hotmail.it (S.C.); laura.cricenti96@gmail.com (L.C.); ilaria.bastoni@policlinicogemelli.it (I.B.); paolaemilia.ferrara@policlinicogemelli.it (P.E.F.); gianpaolo.ronconi@policlinicogemelli.it (G.R.); 3Department of Woman and Child Health and Public Health, Università Cattolica del Sacro Cuore, 00168 Rome, Italy

**Keywords:** respiratory physiotherapy, preterm infants, spontaneous respiratory activity, mechanical ventilation, patent ductus arteriosus

## Abstract

Background: Tactile stimulation manoeuvres stimulate spontaneous breathing in preterm newborns. The aim of this study is to evaluate the effect of early respiratory physiotherapy on the need for mechanical ventilation during the first week of life in preterm infants with respiratory failure. Methods: This is a monocentric, randomised controlled trial. Preterm infants (gestational age ≤ 30 weeks) not intubated in the delivery room and requiring non-invasive respiratory support at birth were eligible for the study. The intervention group received early respiratory physiotherapy, while the control group received only a daily physiotherapy program (i.e., modifying the infant’s posture in accordance with the patient’s needs). Results: between October 2019 and March 2021, 133 preterm infants were studied, 68 infants in the study group and 65 in routine care. The study group showed a reduction in the need for mechanical ventilation (not statistically significant) and a statistically significant reduction in hemodynamically significant patent ductus arteriosus with respect to the control group (19/68 (28%) vs. 35/65 (54%), respectively, *p* = 0.03). Conclusions: early respiratory physiotherapy in preterm infants requiring non-invasive respiratory support at birth is safe and has proven to be protective against haemodynamically significant PDA.

## 1. Introduction

Tactile stimulation manoeuvres are effective techniques for stimulating spontaneous breathing and are recommended for preterm infants already in the delivery room, starting from the first minutes of life [1,2]. However, there is currently no guidance on how, when, and for how long time to perform them to achieve sustained stimulation of spontaneous breathing. In recent years, non-invasive respiratory support has been increasingly used in the treatment of respiratory failure in preterm infants [1,2]. Thus, stimulation of spontaneous breathing represents a crucial point in treatment to avoid intubation and subsequent mechanical ventilation (MV). Currently, the indications of the most recent European Consensus Guidelines on the management of respiratory distress syndrome (RDS) include the use of continuous positive airway pressure (CPAP) and the early administration of caffeine to promote and maintain adequate functional residual capacity (FRC) [2]. 

It has been observed that repeated tactile stimulation in preterm infants leads to a reduction in respiratory effort and improved oxygenation. In particular, Dekker et al. observed that stimulation of the soles of the feet activates propioceptors and stimulation of the newborn’s back stimulates somatic/visceral mechanoreceptors in the thorax, which contribute to stimulation of spontaneous breathing [3]. These afferent somatosensory pathways are functional even before 25 weeks of gestation [4,5]. However, there are no randomised clinical trials investigating the effectiveness of early airway physiotherapy treatment in preterm infants in the first 24 h of life.

The aim of this study was to find out whether early respiratory physiotherapy reduces the need for intubation and MV in the first week of life (excluding the transient tracheal intubation for surfactant administration) in neonates with respiratory failure not intubated in the delivery room and receiving non-invasive respiratory support, by stimulating spontaneous breathing and promoting its maintenance. 

## 2. Materials and Methods

### 2.1. Study Design and Participants

This is an unblinded, monocentric, randomised trial conducted in our third-level neonatal intensive care unit (NICU) between October 2019 and March 2021.

Preterm infants with a gestational age (GA) ≤ 30 weeks not intubated in the delivery room and requiring non-invasive respiratory support at birth due to respiratory distress were eligible.

Exclusion criteria included the presence of major malformations, genetic syndromes, inborn errors of metabolism, foetal hydrops, birth in other hospitals, pulmonary hypertension, severe circulatory disturbances (diagnosed by prolonged capillary refill time, decreased peripheral pulse width, cold skin, lethargy, hypotension, oliguria, increased lactate concentration, and metabolic acidosis), other conditions at the time of randomisation of extreme instability that prevented the implementation of physiotherapy required for the study.

The study protocol was approved by the Ethics Committee of the Fondazione Policlinico Universitario A. Gemelli IRCCS, Rome, Italy, with the approval number 9108/18 ID: 1906 on 26 March 2018. Written and oral information was offered to parents if their infant was likely to be eligible. Informed written consent was obtained from both parents and sufficient time was provided for consent [6]. 

### 2.2. Procedures

All preterm infants admitted to our NICU routinely undergo a physiotherapy program (the Individualized Postural Care Program), which consists of modifying the infant’s posture and decubitus in accordance with the needs of the particular preterm infant [7,8,9]. Postures like prone, supine, and lateral positions, with appropriate support to ensure safety and effectiveness, are changed every 24 h. Manipulations and postures are performed by both specialised nurses and physiotherapists to ensure stability of oxygen saturation, bone integrity, postural control, and sensory and motor stimulation [1].

Preterm infants who met the eligibility criteria were randomly divided into two groups during the first 24 h of life [6].

Infants were randomly assigned in a 1:1 ratio to either a breathing facilitation technique (intervention group) or individual postural care (control group). Ralloc, a Stata/IC 15.1 for Windows (Stata-Corp, College Station, TX, USA), was used to create a sequence of treatments that were randomly permuted into blocks of varying size and order. Concealment of allocation was ensured by sequentially numbered, opaque, sealed envelopes [6].

In the study group, preterm infants received the technique of respiratory facilitation according to the reflex stimulations performed by the physiotherapist [10]. The newborn was placed in supine decubitus and a slight digital pressure was exerted on a hemithorax, more precisely between the 7th and the 8th rib (area corresponding to the insertion of the diaphragm muscle) at the level of the mammillary line, pressing from top to bottom (in the direction of the support plane) and obliquely (in the direction of the vertebral column). This is defined as the “trigger point”. Stimulating the trigger point will stimulate respiratory activity by inducing a compression on the stimulated side with consequent increase in the ipsilateral pulmonary ventilation/minute and the facilitation of the contralateral pulmonary expansion (thoracic expansion of the ribcage). The respiratory facilitation technique was performed for about three minutes and repeated for a total of 4/6 times in sequence: 7th right hemitorax space for three minutes, 7th left hemithorax space for three minutes, 7th right space for a further three minutes, 7th left space for a further three minutes. In case of secretions, the respiratory facilitation technique was associated with the prolonged slow expired technique for the preterm infant. This program was performed 3 times a day until complete respiratory autonomy (absence of any kind of respiratory support and oxygen) was achieved.

In the control group, preterm infants exclusively underwent the Individualized Postural Care program. Physiotherapists performed the technique of respiratory facilitation and autogenous drainage modified only in the presence of clinically and radiographically established pulmonary atelectasis.

Patients of both groups followed the Individualized Family-Centered Developmental Care Program. In particular, attention was focused on the control of environmental stimuli, such as soft lights during treatment to allow the baby to stay awake without being bothered by bright lights directly on the eyes, and reduction of environmental noise (tone of the voice of operators, noise of the monitors, cell phones).

All the interventions of the Individualized Family-Centered Developmental Care Program are in fact aimed at preventing and containing destabilisation and neonatal discomfort considering the extreme sensitivity and vulnerability of these patients to environmental stimulation. In all newborns, the discomfort/pain measurement was performed, using the Neonatal Pain, Agitation, and Sedation Scale (NPASS), according to our NICU protocol [11].

All patients received a loading dose of intravenous caffeine citrate (20 mg/kg) immediately after admission to the NICU, followed by a daily maintenance intravenous dose of 5–10 mg/kg.

The intervention in the study group was also continued in case of intubation and the start of MV during the entire hospital stay, to enhance respiratory function and favour extubation. 

Preterm infants of both groups affected by respiratory distress syndrome (RDS) received surfactant treatment (200 mg/kg of poractant alfa—Chiesi Farmaceutici, Parma, Italy) if they needed FiO_2_ ≥ 0.30 and CPAP of 6–7 cm H_2_O, by the IN-SUR-E technique (GA ≥ 28 weeks) or IN-REC-SUR-E technique (GA < 28 weeks), as per our NICU protocol [12]. 

The indications for MV were poor oxygenation with FiO_2_ greater than 0.40 after rescue surfactant, respiratory acidosis (pCO_2_ > 65 mm Hg [8.5 kPa] and pH < 7.20), or apnoea (more than four episodes of apnoea per hour or more than two episodes of apnoea per hour requiring ventilation with bag and mask), despite optimal non-invasive respiratory support (nasal CPAP, nasal intermittent positive pressure ventilation, or bilevel positive airway pressure) [6].

### 2.3. Data Collection and Outcomes

Demographic data on patient and maternal characteristics were collected from each patient. 

The following parameters were continuously recorded: Arterial Oxygen Saturation (SpO_2_), Heart Rate (HR), Respiratory Rate (RR), Fraction of Inspired Oxygen (FiO_2_), nCPAP level, episodes of apnoea defined as cessation of breathing for more than 20 s, or a shorter respiratory pause associated with oxygen desaturation and/or bradycardia, episodes of bradycardia (HR < 80 bpm) or tachycardia (HR > 180 bpm), presence of secretions, NPASS Score, and Silverman Anderson Score.

At the start (T0) and at the end (T1) of the reflex stimulations in the study group and at the same time in the control group, HR, RR, SpO_2_, and FiO_2_ were recorded. 

The mean ± SD of the values obtained for each parameter during the 3 daily stimulating procedures was used for the final analysis.

Non-invasive pulse oximetry SpO_2_/FiO_2_ ratio (SFR) was evaluated. The SFR was the ratio between peripheral oxygen saturation by pulsoximetry (measured by Masimo SET technology) and FiO_2_, as indicated by ventilator or CPAP devices. The SFR value has demonstrated its effectiveness in providing a reliable indication of oxygenation levels in preterm infants, similar to the P/F ratio (arterial oxygen concentration (PaO_2_) to inspired oxygen concentration (FiO_2_) ratio) in a variety of clinical settings, including pulmonary involvement during sepsis, anaesthesia, and surfactant administration [13].

The SFR value in the first 24 h of life was calculated at T0 before treatment and at T1 at the end of treatment. On subsequent days, SFR value per day was averaged for the first 7 days of treatment. 

Duration of MV, non-invasive respiratory support, and O_2_-therapy in the NICU and total length of hospital stay were recorded for each patient.

The primary outcome of this study was the need for intubation and MV in the first week of life [6].

The secondary outcomes were as follows: duration of mechanical ventilation during the hospital stay;duration of non-invasive respiratory support during the hospital stay;duration of O_2_-therapy during the hospital stay;occurrence of bronchopulmonary dysplasia (BPD) according to the 2001 National Institutes of Health (NIH) consensus definition [14];pulmonary atelectasis diagnosed on chest X-ray and/or lung ultrasound;length of hospital stay (LOS);survival.

The further outcomes of this study were to find out whether the consequences of preterm birth can be influenced (increased or decreased) in any way by early respiratory physiotherapy. Additional data recorded for each infant then included the occurrence of the following:haemodynamically significant patent ductus arteriosus according to ultrasound criteria that include PDA characteristics such as ductal size and ductal flow, overload indices such as LVO, LA/Ao ratio, and indices of systemic shunt impairment (flow pattern in anterior cerebral artery or superior mesenteric artery) [15];grade 3–4 intraventricular haemorrhage (IVH) [16];periventricular leukomalacia [17];sepsis, defined as a positive blood culture or suggestive clinical and laboratory findings leading to treatment with antibiotics for at least 7 days despite absence of a positive blood culture;pneumothorax;retinopathy of prematurity (ROP) [18];pulmonary haemorrhage.

### 2.4. Statistical Analysis

We calculated the study sample size based on analysis of our 2017 NICU data. The incidence of intubation and mechanical ventilation in the first week of life in preterm infants with GA ≤ 30 weeks who were not intubated in the delivery room was 50%. The expected effect of the intervention proposed in the study was an absolute risk reduction of 25% (relative risk reduction of 50%) from the expected control group rate. Allowing for an α-error of 0.05 (two-sided testing) and a study power of 80%, 66 newborns per arm needed to be included (132 newborns in total); this number was increased by 10% to take account of any dropouts. 

Statistical analysis was carried out using the “intention to treat” method; all data were collected in a single database and analysed to evaluate any differences between the randomised groups both for primary outcome and for secondary outcomes. 

Data were reported as mean and standard deviation or median (range) and compared using Mann–Whitney U test or Student t-test as appropriate. Categorical variables were expressed as numbers and percentages and compared using Fisher’s exact test. Statistical significance was set for *p*-values < 0.05. Statistical analysis was performed with the Stata Statistical Software: Release 18.0 software (StataCorp LP, College Station, TX, USA).

This study is registered with UMIN-CTR Clinical Trial UMIN000036066.

## 3. Results

Between October 2019 and March 2021, a total of 153 elegible preterm infants were born; of these, 20 could not be included in the study because their parents refused to give consent. Of the remaining 133 elegible preterm infants, 68 were included in the study group and 65 in the control group (Figure 1). 

### 3.1. Results about Demographic Data and Clinical Characteristics

No significant differences were observed between the two groups in terms of principal demographic data (patient and maternal characteristics) and clinical characteristics, as shown in Table 1.

### 3.2. Results about Outocomes

No statistically significant difference was found with regard to the primary outcome (the need for MV in the first week of life) between the two groups, although there was a marked downward trend in the study group (4% vs. 9%; *p* = 0.32), as shown in Table 2.

No statistically significant differences were found between the two groups regarding the need for MV after the first week of life either. Considering only patients who required MV, a decreasing trend was observed in the study group, which received a total median length of MV of 54 h, compared to 168 h in the control group (*p* = 0.16, Table 2). 

A total of 18 studied patients, 10 (15%) in the study group and 8 (12%) in the control group, did not require non-invasive support and oxygen supplementation during their hospital stay (*p* = 0.8). The median values of non-invasive respiratory support in the remaining 115 patients were 17 days in the study group vs. 22 days in the control group (*p* = 0.48) (Table 2).

The surfactant was administered in 27 (40%) preterm infants in the study group, and it was administered in 27 (41%) preterm infants as well in the control group. No statistically significant difference was found in the Heart Rate (HR), Respiratory Rate (RR), and episodes of apnoea between the two groups.

No statistically significant difference was found in the duration of oxygen therapy in the 115 studied patients requiring it; the median values were 14 days in the study group vs. 21 days in the control group (*p* = 0.26, Table 2).

No statistically significant difference was found in the discomfort or pain, measured using the Neonatal Pain, Agitation, and Sedation Scale (NPASS) between the two groups.

No statistically significant difference was found in the two groups about the incidence of BPD.

The incidence of hsPDA was 28% in the study group and 54% in the control group (*p* = 0.03), as shown in Table 2. The type of drug used was not the subject of the study, but in the three-year period studied, the drug used in the first treatment was acetaminophen in 46%. The first day of treatment lasted an average of 10 ± 8 days, and the number of cycles required for completion was 1.8 ± 0.9. 

No statistically significant difference was found regarding the incidence of severe IVH (>2 grade according to Papile), although there was a downward trend in the study group (4 (6%) cases vs. 8 (12%) cases in the control group) (*p* = 0.24) (Table 2).

There were no statistically significant differences between the two groups in terms of length of stay, periventricular leukomalacia, incidence of sepsis, and mortality (Table 2). 

No cases of pulmonary atelectasis, pneumothorax, or pulmonary haemorrhage occurred in either group. 

The SpO_2_/FiO_2_ ratio before the first physiotherapy treatment (T0, in the first 24 h of life) was 426 ± 51 in the study group, while it was 420 ± 60 in the control group (*p* = 0.8). At the end of the first respiratory physiotherapy treatment (T1), this ratio was 432 ± 51 in the study group and 418 ± 63 in the control group (*p* = 0.30), proving not statistically significant, but showing a tendency towards higher values.

The SFR value calculated daily over the next seven days of life showed no statistically significant differences between the two groups, although there was an upward trend in the SpO_2_/FiO_2_ value in the study group compared to the control group, for every single day considered (Table 3). Furthermore, there was an increasing trend in the SpO_2_/FiO_2_ value from day 1 to day 7 of life in the patients of the study group (Table 3).

The CPAP values calculated daily in the two groups showed no statistically significant differences, although there was a downward trend in the study group, for every single day considered (Table 3). 

## 4. Discussion

Respiratory physiotherapy is a rare practice in neonatal intensive care units, both because of the need for skilled personnel and the extreme complexity of the patients. Treatment of preterm infants in the first days of life is not always advocated in the literature, as there is evidence that the infant’s posture may favour the occurrence of intraventricular haemorrhage [19]. However, the child’s posture is quite different from tactile stimulation and the more specific respiratory physiotherapy, which is carried out by experienced and dedicated staff and according to very specific rules [20,21].

The first aim of our study was to highlight the safety of physiotherapy manoeuvres performed by experienced staff. The results confirm that respiratory physiotherapy is a safe procedure that does not harm and can even be beneficial in stabilizing the patient, especially during the most delicate days of his life.

The primary endpoint was not met, although a reduction was observed. Thus, the main objective must be confirmed by further studies. A similar reduction was also found in terms of total length of MV, non-invasive respiratory support, and oxygen therapy.

It is interesting to note how, in both groups, there was a considerable reduction in the incidence of need for MV (4% in the study group and 9% in the control group), compared to the expected incidence of 50% found in our population prior to the study (year 2017), on which we based our sample size. Reducing MV demand in preterm infants is a goal we have been working toward for years, including with the nursing staff. In these years, especially from 2018, we have increased the use of non-invasive respiratory support for the management of preterm infants with GA ≤ 30 weeks, in line with international guidelines, [2,22], thanks also to the possibility of having more and more powerful non-invasive ventilation devices and thanks to the focus on extubation as early as possible, with a strong emphasis on physiotherapy and nursing care. The downward trend had already started before 2020, so COVID-19 may have contributed, but we believe it is something we have worked hard towards regardless of the pandemic.

Several studies show that tactile stimulation promotes spontaneous breathing activity by changing the arousal state. However, most of these stimulations are performed in the delivery room during the first minutes of life in apnoeic infants as part of resuscitation measures [21]. There was no difference found in apnoea in either group. For this reason, we believe that respiratory physiotherapy also affects stabilisation through other mechanisms; our data show that infants in the study group required less PEEP, although not in a statistically significant way. It is possible that physiotherapy leads to a better thoracic expansion and to a lower tendency toward respiratory collapse by stimulating trigger points.

A very interesting finding is that in the group of infants receiving early respiratory physiotherapy, spontaneous closure of the ductus arteriosus occurred significantly more often without having to resort to drug therapies, which, although effective, are nevertheless fraught with numerous side effects that can be potentially fatal for our little patients. According to our internal protocols, to screen for PDA in the at-risk population (i.e., infants of both arms, infants with the same demographic characteristics), an initial functional echocardiogram is performed between 48 and 72 h after birth. If hsPDA are present at first detection, an ultrasound scan is performed after 24–48 h, and if the significance criteria are confirmed, pharmacological treatment is initiated, with ibuprofen as the first choice, unless there are claimed contraindications, such as ongoing acute kidney injury. The performance of physiotherapeutic manoeuvres did not change the echographic assessment.

The pathophysiological mechanism underlying this finding may be the increase in paO_2_, which physiologically favours spontaneous closure of the ductus arteriosus. From the studies performed in the delivery room, it appears that tactile stimulation leads to an increase in paO_2_ in preterm infants [21]. The same could happen if these stimulations are performed repeatedly and regularly, as in our study, precisely during the critical days for spontaneous closure of the ductus arteriosus. In this sense, the higher SpO_2_/FiO_2_ ratio after physiotherapy treatment (T1) in the patients of the study group compared with the control group confirms this hypothesis. We do not believe that the incidence of hsPDA can be influenced by the difference in the technique used for surfactant administration (IN-SUR-E vs. IN-REC-SUR-E) because the populations are different. In our NICU, IN-REC-SUR-E is more standardly performed in neonates with GA < 28 weeks, thus a different group compared to those undergoing physiotherapy (GA ≤ 30 weeks). Therefore, the differences in incidence could be justified by the differences in GA, one of the main risk factors for the presence of hsPDA.

The lower incidence of PDA in the study group is of interest; however, it should be clarified that the study was not methodologically designed for this purpose. Therefore, we believe that these data are interesting but need to be confirmed by other studies with a larger sample.

Spontaneous closure of the duct also does not seem to be promoted by the presence of higher CPAP levels; in the study group, the SpO_2_/FiO_2_ ratio appeared to be higher, although CPAP levels tended to be lower. This could indicate that respiratory physiotherapy resulted in greater stability in the infants being treated, regardless of the CPAP level.

Another interesting finding is the trend towards reduction in the incidence of severe IVH (>2 grade according to Papile) in newborns treated with early respiratory physiotherapy with respect to the control group. The hypothesis supporting this finding is that tactile stimulation maintains more stable cerebral blood flow, as shown by Bassani et al. [20]. 

## 5. Conclusions

The data from this study, to our knowledge the first randomised controlled trial examining the effects of early respiratory physiotherapy in preterm infants (GA ≤ 30 weeks) in the setting of non-invasive respiratory support, show that early respiratory physiotherapy in this category of preterm infants is safe, does not expose them to a greater incidence of short- and long-term respiratory sequelae, and has proven to be protective against the occurrence of haemodynamically significant PDA.

In our unit, respiratory physiotherapy is, in fact, performed from the very first day of life in at-risk infants (GA ≤ 30 weeks), because we have been reassured by our results and strongly believe in the benefits of early manipulations performed by experienced staff. 

Further studies, on a larger sample, could be useful for confirming our findings and also finding future research directions.

## Figures and Tables

**Figure 1 children-10-01761-f001:**
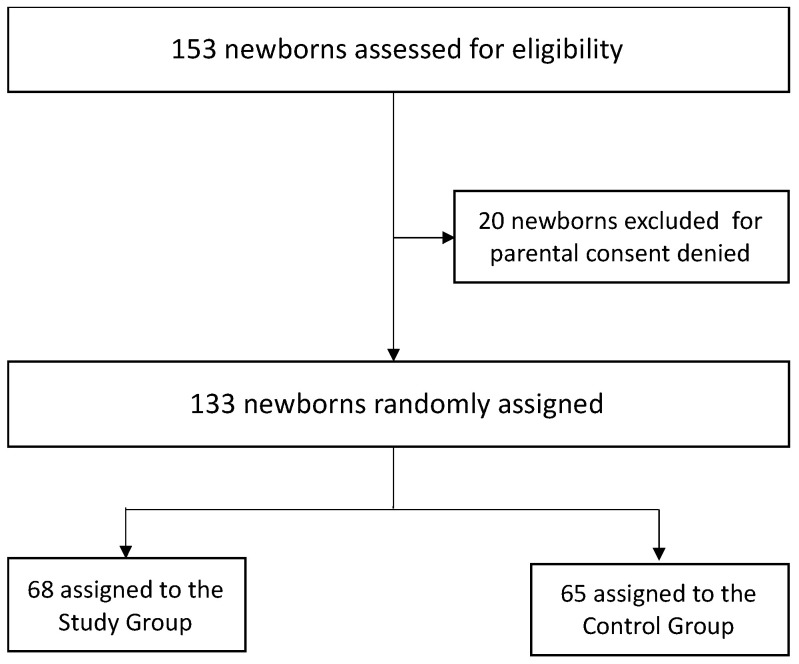
Infant flow diagram during the study period.

**Table 1 children-10-01761-t001:** Principal demographic data and clinical characteristics.

Characteristics	Study Group *n* = 68 (51%)	Control Group *n* = 65 (49%)	*p*-Value
Gestational Age (weeks)	28.3 ± 1.6	28.0 ± 1.9	0.29
Birth Weight (g)	1040 ± 270	1044 ± 325	0.93
Male	32 (47)	30 (46)	0.49
Complete course of antenatal steroids * *n* (%)	31 (46)	33 (51)	0.6

Values expressed as mean ± SD and number (per cent). * A complete course of antenatal steroids was defined as two doses of betamethasone administered more than 24 h but no more than 7 days before delivery.

**Table 2 children-10-01761-t002:** Effect of early respiratory physiotherapy on primary outcome and secondary outcomes.

Outcome	Study Group *n* = 68 (51%)	Control Group *n* = 65 (49%)	*p*-Value
Mechanical Ventilation within the first 7 days of life	3 (4)	6 (9)	0.32
Mechanical Ventilation after the first 7 days of life	3 (4)	2 (3)	1
Total Mechanical Ventilation, hours *	54 [3–1056]	168 [48–1248]	0.16
Total non-invasive respiratory support, days **	17 [1–404]	22 [1–116]	0.48
Total Oxygen Therapy, days ***	14 [1–159]	21 [1–117]	0.26
Bronchopulmonary dysplasia (BPD) ^†^	3 (4)	6 (9)	0.32
Pulmonary atelectasis	0 (0)	0 (0)	1
Mortality	10 (15)	10 (15)	1
Length of hospital stay, days	64 ± 32	69 ± 43	0.40
Intraventricular haemorrhage worse than grade 2	4 (6)	8 (12)	0.24
Periventricular leukomalacia	2 (3)	4 (6)	0.43
hs PDA ^††^	19 (28)	35 (54)	0.003
Sepsis	43 (63)	45 (69)	0.58
Retinopathy of prematurity, any grade	22 (32)	22 (34)	1

Values expressed as number (per cent), mediane (range) and mean ± SD. * Data referred only to studied patients requiring mechanical ventilation. ** Data referred only to studied patients requiring non-invasive respiratory support. *** Data referred only to studied patients requiring oxygen therapy. ^†^ Bronchopulmonary dysplasia (BPD) according to the 2001 National Institutes of Health consensus definition. ^††^ hs PDA = patent ductus arteriosus, haemodynamically significant, requiring pharmacological treatment.

**Table 3 children-10-01761-t003:** SpO_2_/FiO_2_ ratio and CPAP level in the first week of treatment.

Outcome	Study Group *n* = 68 (51%)	Control Group *n* = 65 (49%)	*p*-Value
SpO_2_/FiO_2_ T0 °	426 ± 51	420 ± 60	0.80
SpO_2_/FiO_2_ T1 °°	432 ± 51	418 ± 63	0.30
SpO_2_/FiO_2_ D1	365 ± 92	370 ± 84	0.71
SpO_2_/FiO_2_ D2	422 ± 50	421 ± 60	0.98
SpO_2_/FiO_2_ D3	420 ± 55	418 ± 66	0.81
SpO_2_/FiO_2_ D4	427 ± 60	417 ± 76	0.38
SpO_2_/FiO_2_ D5	430 ± 57	423 ± 62	0.48
SpO_2_/FiO_2_ D6	437 ± 50	430 ± 61	0.47
SpO_2_/FiO_2_ D7	444 ± 43	427 ± 64	0.08
CPAP level D1, cmH_2_O	6.4 ± 0.9	6.4 ± 1.0	0.68
CPAP level D2, cmH_2_O	5.8 ± 1.2	6.0 ± 1.4	0.40
CPAP level D3, cmH_2_O	5.4 ± 1.3	5.8 ± 1.5	0.21
CPAP level D4, cmH_2_O	5.2 ± 1.4	5.7 ± 1.4	0.12
CPAP level D5, cmH_2_O	5.3 ± 1.4	5.6 ± 1.6	0.34
CPAP level D6, cmH_2_O	5.3 ± 1.6	5.4 ± 1.7	0.80
CPAP level D7, cmH_2_O	5.2 ± 1.6	6.0 ± 1.9	0.09

° T0 = Time before physiotherapy treatment in the first 24 h of life. °° T1 = Time after the first physiotherapeutic treatment in the first 24 h of life. CPAP = continuous positive airway pressure. D = day of life.

## Data Availability

The data presented in the present study are available on request from the corresponding author.

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
