# Peer review of "Early Respiratory Physiotherapy versus an Individualized Postural Care Program for Reducing Mechanical Ventilation in Preterm Infants: A Randomised Controlled Trial"

_children, 2023, doi:10.3390/children10111761_

Round 1
Reviewer 1 Report
Comments and Suggestions for Authors
The authors report the results of a non blinded randomized single center trial assessing the utility of early respiratory physiotherapy on the need of mechanical ventilation during the first week of life in preterm infants.
Overall, the study is well conducted with good methodology and the results are presented in a easy to read format. That said, there are some limitations and missing data which I feel should be addressed prior to publication.
Major Limitations:
The study is underpowered to detect a difference in the need for mechanical ventilation. The authors acknowledge this in the discussion section and state them adopting NIPPV and perhaps routine caffeine has resulted in the recent decrease compared to their historical cohort of patients which formed the basis for the power calculation. As such, if the study is underpowered to detect a change in the primary outcome, the authors should refrain from using the term "marked downward trend in the need for mechanical ventilation" and rather perhaps focus on safety. I commend the authors that their need for mechanical ventilation has become so low but this makes detecting a difference in that outcome very difficult and a difference of a N of 3 cannot say much about the impact of this intervention.
The author's had a serendipitous result regarding the the reduction of hemodynamically significant PDA in the intervention arm. They make some guesses as to the physiological basis behind why this may be possible which is sound, however, the authors should also investigate whether other differences in management such as IN-SUR-E vs IN-REC-SUR-E would have also made a difference. In general, more data would be needed prior to attributing this decrease to the intervention:
1. The definition of hemodynamically significant PDA used - was it uniformly applied.
2. The timing of PDA assessment in both arms - was it similar and whether a single assessment or multiple assessment, any temporal relationship between intervention and non detection of PDA.
3 The incidence of PDA therapy in both groups - the kind of therapy and timing.
Lastly, the authors should acknowledge that this study was not powered to detect a difference in hemodynamically significant PDA.
Minor Limitations:
1. The background for the study largely stresses on the impact of stimulation on respiratory drive. However, as the authors note that there was no difference in apnea in both groups and with the use of NIPPV and caffeine, lack of respiratory drive is unlikely to be the reason for mechanical ventilation in this population. Perhaps the authors could highlight other benefits of physiotherapy and positional changes which may have an impact on respiratory dynamics in this population.
2. The primary outcome of Pulmonary atelectasis diagnosed on chest X-ray and/or lung ultrasound has no results reported.
3. Rates of surfactant administration between both groups should be reported.
Comments on the Quality of English LanguageWell written but minor grammatical errors such as in Line 25 - "Tactile stimulation manoeuvres stimulate spontaneous breathing, also in 25 preterm newborns". Is it meant to be "Tactile stimulation manoeuvres stimulate spontaneous breathing in preterm newborns"
Author Response
Major Limitations:
- The study is underpowered to detect a difference in the need for mechanical ventilation. The authors acknowledge this in the discussion section and state them adopting NIPPV and perhaps routine caffeine has resulted in the recent decrease compared to their historical cohort of patients which formed the basis for the power calculation. As such, if the study is underpowered to detect a change in the primary outcome, the authors should refrain from using the term "marked downward trend in the need for mechanical ventilation" and rather perhaps focus on safety. I commend the authors that their need for mechanical ventilation has become so low but this makes detecting a difference in that outcome very difficult and a difference of a N of 3 cannot say much about the impact of this intervention.
Reply: Thank you for your comment. The primary endpoint was not met, although a reduction was observed. Thus, the main objective must be confirmed by further studies. However, we can say that our results show that the procedure, apart from its potential advantages that need to be further confirmed, is safe in such a vulnerable population. We have made this change to the article on line 33 and 312-317.
- The author's had a serendipitous result regarding the reduction of hemodynamically significant PDA in the intervention arm. They make some guesses as to the physiological basis behind why this may be possible which is sound, however, the authors should also investigate whether other differences in management such as IN-SUR-E vs IN-REC-SUR-E would have also made a difference. In general, more data would be needed prior to attributing this decrease to the intervention.
Reply: We do not believe that the incidence of hsPDA can be influenced by the difference in the technique used for surfactant administration (IN-SUR-E vs IN-REC-SUR-E) because the populations are different. In our NICU, IN-REC-SUR-E is more standardized performed in neonates with a GA < 28 weeks, thus a different group compared to those undergoing physiotherapy (GA ≤ 30 weeks). Therefore, the differences in incidence could be justified by the differences in GA, one of the main risk factors for the presence of hsPDA. We have made this change to the article on line 361-367.
- The definition of hemodynamically significant PDA used - was it uniformly applied.
Reply: Thank you for your comment. We have used internal protocols for screening and treatment of hsPDA. The current internal protocols are the result of the best evidence in the literature, although we are aware of the difficulty in standardizing decisions about the treatment of hsPDA. It must be said, however, that the significance criteria used, the ultrasound criteria, are the most widely recognized by the leading experts in the field. They are uniformly applied to infants at highest risk for hsPDA, ie, preterm infants with a gestational age of less than 30 weeks.
Criteria for hemodynamic significance are ultrasound criteria that include PDA characteristics such as ductal size and ductal flow, overload indices such as LVO, LA /Ao ratio, and indices of systemic shunt impairment (flow pattern in Anterior cerebral artery or Superior mesenteric artery). We have reported these criteria in the article in line 188-191.
- The timing of PDA assessment in both arms - was it similar and whether a single assessment or multiple assessment, any temporal relationship between intervention and non-detection of PDA.
Reply: According to our internal protocols, to screen for PDA in at-risk population (ie, infants of both arms, infants with the same demographic characteristics), an initial functional echocardiogram is performed between 48 and 72 hours after birth. Further follow-up examinations are based on the results of the first echocardiogram. The performance of physiotherapeutic manoeuvres did not change the echographic assessment. We have made this change to the article on line 346-348 and 352-353.
- The incidence of PDA therapy in both groups - the kind of therapy and timing.
Reply: The incidence of PDA therapy was 28% in the study group and 54% in the control group, as shown in Table 2. The type of drug used was not the subject of the study, but in the three-year period studied, the drug used in the first treatment was ibuprofen in 54%. The first day of treatment lasted an average of 10 ± 8 days, and the number of cycles required for completion was 1.8 ± 0.9. We have made this change to the article in line 270-275.
- Lastly, the authors should acknowledge that this study was not powered to detect a difference in hemodynamically significant PDA.
Reply: The lower incidence of PDA in the study group is of interest; however, it should be clarified that the study was not methodologically designed for this purpose. Therefore, we believe that these data are interesting but need to be confirmed by other studies with a larger sample. We have made this change to the article on line 368-371.
Minor Limitations:
- The background for the study largely stresses on the impact of stimulation on respiratory drive. However, as the authors note that there was no difference in apnea in both groups and with the use of NIPPV and caffeine, lack of respiratory drive is unlikely to be the reason for mechanical ventilation in this population. Perhaps the authors could highlight other benefits of physiotherapy and positional changes which may have an impact on respiratory dynamics in this population.
Reply: Thank you for your comment. We agree that respiratory physiotherapy also affects stabilization through other mechanisms; our data show that infants in the study group required less PEEP, although not in a statistically significant way; it is possible that physiotherapy leads to a better thoracic expansion and to a lower tendency to respiratory collapse by stimulating trigger points. We have made this change to the article on line 337-341.
- The primary outcome of Pulmonary atelectasis diagnosed on chest X-ray and/or lung ultrasound has no results reported.
Reply: Thank you for your comment. Pulmonary atelectasis was not diagnosed on chest X-ray and/or lung ultrasound in both group as we had written in the sentence in the text on line 281. We have inserted this data on Table 2.
- Rates of surfactant administration between both groups should be reported.
Reply: Thank you for your comment. The surfactant was administered in 27 (40%) preterm infants in the study group (and it was administered in 27 (41%) preterm infants as well in the control group. We added this sentence in the text on line 259-260.
Comments on the Quality of English Language
Well written but minor grammatical errors such as in Line 25 - "Tactile stimulation manoeuvres stimulate spontaneous breathing, also in 25 preterm newborns". Is it meant to be "Tactile stimulation manoeuvres stimulate spontaneous breathing in preterm newborns"
Reply: Correction done on line 24.
Reviewer 2 Report
Comments and Suggestions for Authors
Dear colleagues thank you for the exceptional research submitted. It was a real pleasure to review such a great paper.
As you stated in Material and methods section, page 5 "We calculated the study sample size based on analysis of our 2017 NICU data. The incidence of intubation and mechanical ventilation in the first week of life in preterm infants with GA ≤ 30 weeks who were not intubated in the delivery room was 50%. The expected effect of the intervention proposed in the study was an absolute risk reduction of 25% (relative risk reduction of 50%) from the expected control group rate. Allowing for an α-error of 0.05 (two-sided testing) and a study power of 80%, 66 newborns per arm needed to be included (132 newborns in total); this number was increased by 10% to take account of any dropouts." In Discussion section, page 8, you are stating "It is interesting to note how in both groups there was a considerable reduction in the incidence of need for MV (4% in the study group and 9% in the control group), compared to the expected incidence of 50% found in our population prior to the study (year 2017), on which we based our sample size."
Can you comment in more detail on causes of this rather spectacular reduction in MV need that was documented in the premature patients in such a short time?
Could pandemic non-pharmaceutical measures and extreme precautions associated with COVID-19 disease be involved in this reduction from 50% to 9% in control group? Was the staff involved in care for these children more proactive in infection-control or other measures were implemented in your unit as compared with pre-pandemic era?
It would strengthen the scientific impact of your great research if you will provide in Discussion section some data on MV need in 2018 and 2019, in your department, to corroborate with your explanation of this phenomenon "A plausible explanation is that over these years, particularly starting from 2018, we have strengthened the use of non invasive respiratory support for the management of preterm infants of GA ≤ 30 weeks, in line with international guidelines".
You are presenting very convincing data on SpO2/FiO2 ratio and CPAP values during study. Do you have any data on lung ultrasound monitoring of these children?
Because you have proven in a previously published paper that LUS trajectories correlates with oxygen status and could predict BPD [Raimondi F, Migliaro F, Corsini I, Meneghin F, Dolce P, Pierri L, Perri A, Aversa S, Nobile S, Lama S, Varano S, Savoia M, Gatto S, Leonardi V, Capasso L, Carnielli VP, Mosca F, Dani C, Vento G, Lista G. Lung Ultrasound Score Progress in Neonatal Respiratory Distress Syndrome. Pediatrics. 2021 Apr;147(4):e2020030528. doi: 10.1542/peds.2020-030528.]
Regarding circulatory stability you are stating that "The incidence of PDAhs was significantly lower in the study group (p =0.03) (Table 2).". Please elaborate more on the evaluation protocol for determining hemodynamic instability PDA-induced in these children.
It would also help to insert briefly some data on how was this PDA hemodynamic instability managed in these scenarios.
Did these finding generate any change in protocols implemented in your unit since study concluded in March 2021? I am focusing on a potential rephrasing of your statement in Conclusion section, page 9, that could increase the impact of these findings: "The described technique of respiratory facilitation according to the reflex stimulations performed by physiotherapist should be implemented in NICUs, already in the first 24 hours of life"
Please unify the citation style in references 15-18 that have a complete different citation aspect compared with all other papers cited.
Comments on the Quality of English LanguageNo significant issues documented
Author Response
- As you stated in Material and methods section, page 5 "We calculated the study sample size based on analysis of our 2017 NICU data. The incidence of intubation and mechanical ventilation in the first week of life in preterm infants with GA ≤ 30 weeks who were not intubated in the delivery room was 50%. The expected effect of the intervention proposed in the study was an absolute risk reduction of 25% (relative risk reduction of 50%) from the expected control group rate. Allowing for an α-error of 0.05 (two-sided testing) and a study power of 80%, 66 newborns per arm needed to be included (132 newborns in total); this number was increased by 10% to take account of any dropouts." In Discussion section, page 8, you are stating "It is interesting to note how in both groups there was a considerable reduction in the incidence of need for MV (4% in the study group and 9% in the control group), compared to the expected incidence of 50% found in our population prior to the study (year 2017), on which we based our sample size."
Can you comment in more detail on causes of this rather spectacular reduction in MV need that was documented in the premature patients in such a short time?
Could pandemic non-pharmaceutical measures and extreme precautions associated with COVID-19 disease be involved in this reduction from 50% to 9% in control group? Was the staff involved in care for these children more proactive in infection-control or other measures were implemented in your unit as compared with pre-pandemic era? It would strengthen the scientific impact of your great research if you will provide in Discussion section some data on MV need in 2018 and 2019, in your department, to corroborate with your explanation of this phenomenon "A plausible explanation is that over these years, particularly starting from 2018, we have strengthened the use of non invasive respiratory support for the management of preterm infants of GA ≤ 30 weeks, in line with international guidelines".
Reply: Thank you for your comment. Reducing MV demand in preterm infants is a goal we have been working toward for years, including with the nursing staff. In these years, especially from 2018, we have increased the use of noninvasive respiratory support for the management of preterm infants with GA ≤ 30 weeks, in line with international guidelines, thanks also to the possibility of having more and more powerful noninvasive ventilation devices and thanks to the focus on extubation as early as possible, with a strong emphasis on physiotherapy and nursing care. The downward trend had already started before 2020, so Covid-19 disease may have contributed, but we believe it is something we have worked hard towards regardless of the pandemic. We added this sentence in the text on line 325-333.
- You are presenting very convincing data on SpO2/FiO2 ratio and CPAP values during study. Do you have any data on lung ultrasound monitoring of these children?
Reply: Thank you for your comment. At the time of the study, ultrasound monitoring was still unsystematic and experimental; therefore, the data we collected in those years are not usable for the purpose required by the reviewer. However, the use of chest ultrasound in our unit has become a standard and routine practice in the diagnostic evaluation of both infants with GA ≤ 30 weeks and infants with higher gestational age and respiratory disease. The data collected in the last two years are also promising in terms of predicting the development of BPD.
- Regarding circulatory stability you are stating that "The incidence of hsPDA was significantly lower in the study group (p =0.03) (Table 2).". Please elaborate more on the evaluation protocol for determining hemodynamic instability PDA-induced in these children.It would also help to insert briefly some data on how was this PDA hemodynamic instability managed in these scenarios.
Reply: Thank you for your comment. According to our internal protocols, to screen for PDA in the at-risk population (ie, infants of both arms, infants with the same demographic characteristics), an initial functional echocardiogram is performed between 48 and 72 hours after birth. Criteria for hemodynamic significance are ultrasound criteria that include PDA characteristics such as ductal size and ductal flow, overload indices such as LVO, LA /Ao ratio, and indices of systemic shunt impairment (flow pattern in Anterior cerebral artery or Superior mesenteric artery). If hsPDA are present at first detection, an ultrasound scan is performed after 24-48 hours, and if the significance criteria are confirmed, pharmacological treatment is initiated, with ibuprofen as the first choice, unless there are claimed contraindications, such as ongoing acute kidney injury. We added this sentence in the text on line 346-351 and 188-191.
- Did these finding generate any change in protocols implemented in your unit since study concluded in March 2021? I am focusing on a potential rephrasing of your statement in Conclusion section, page 9, that could increase the impact of these findings: "The described technique of respiratory facilitation according to the reflex stimulations performed by physiotherapist should be implemented in NICUs, already in the first 24 hours of life."
Reply: Thank you for the careful clarification. In our unit, respiratory physiotherapy is, in fact, performed from the first day of life in at-risk infants (GA ≤ 30 weeks), because we have been reassured by our results and strongly believe in the benefits of early manipulations performed by experienced staff. We added this sentence in the text in line 387-390.
- Please unify the citation style in references 15-18 that have a completely different citation aspect compared with all other papers cited.
Reply: Correction done.
Round 2
Reviewer 1 Report
Comments and Suggestions for Authors
I congratulate the authors on their study and hard work and it was my privilege to be able to review their work.
I appreciate the changes the authors made, I believe it helps with the generalizability of the of the study and gives more insight into the practice at the author's institution. I would suggest one last minor change:
Since the authors acknowledge that the finding of reduction of hsPDA required further study to validate as it was not the primary outcome, I would suggest that we change the wording in line 38 from "has proven to be protective against haemodynamically significant PDA" to "may be protective against haemodynamically significant PDA"